# SyntheOcc: Synthesize Geometric-Controlled Street View Images through 3D Semantic MPIs

## Abstract

The advancement of autonomous driving is increasingly reliant on high-quality annotated datasets, especially in the task of 3D occupancy prediction, where the occupancy labels require dense 3D annotation with significant human effort. In this paper, we propose **SytheOcc**, which denotes a diffusion model that Synthesize photorealistic and geometric-controlled images by conditioning Occupancy labels in driving scenarios. This yields an unlimited amount of diverse, annotated, and controllable datasets for applications like training perception models and simulation. SyntheOcc addresses the critical challenge of how to efficiently encode 3D geometric information as conditional input to a 2D diffusion model. Our approach innovatively incorporates 3D semantic multi-plane images (MPIs) to provide comprehensive and spatially aligned 3D scene descriptions for conditioning. As a result, SyntheOcc can generate photorealistic multi-view images and videos that faithfully align with the given geometric labels (semantics in 3D voxel space). Extensive qualitative and quantitative evaluations of SyntheOcc on the nuScenes dataset prove its effectiveness in generating controllable occupancy datasets that serve as an effective data augmentation to perception models.

## 1 Introduction

With the rapid development of generative models, they have shown realistic image synthesis and diverse controllability. This progress has opened up new avenues for dataset generation in autonomous driving (Gao et al., 2024; Swerdlow et al., 2024; Wen et al., 2023; Li et al., 2023a). The task of dataset generation is usually modeled as controllable image generation, where the ground truth (*e.g.* 3D Box) is employed to control the generation of new datasets in downstream tasks (*e.g.* 3D detection). This approach helps to mitigate the data collection and annotation effort as it can generate labeled data for free. However, a novel task of vital importance, occupancy prediction (Wang et al., 2023b; Tian et al., 2024), poses new challenges for dataset generation compared with 3D detection. It requires finer and more nuanced geometry controllability, which refers to use the occupancy state and semantics of voxels in the whole 3D space to control the image generation. We argue that solving this problem not only allows us to synthesize occupancy datasets, but also empowers valuable applications such as editing geometry to generate rare data for corner case evaluation, as shown in Fig. 1. In the following, we first illustrate why prior work struggles to achieve the above objective, and then demonstrate how we address these challenges.

In the area of diffusion models, several representative works have displayed high-quality image synthesis; however, they are constrained by limited 3D controllability: they are incapable of editing 3D voxels for precise control. For example, BEVGen (Swerdlow et al., 2024) generates street view images by conditioning BEV layouts using diffusion models. MagicDrive (Gao et al., 2024) extend BEVGen and additionally converts the 3D box parameters into text embedding through Fourier mapping that is similar to NeRF (Mildenhall et al., 2020), and uses cross-attention to learn conditional generation. Although these methods achieve satisfactory results in image generation, their 3D controllability is inherently limited. These approaches are restricted to manipulating the scene in types of 3D boxes and BEV layouts, and hardly adapt to finer geometry control such as editing the shape of objects and scenes. Meanwhile, they usually convert conditional input into 1D embedding that aligns with prompt embedding, which is less effective in 3D-aware generation due to lack of spatial alignment with the generated images. This limitation hinders their utility in downstream

Figure 1: A showcase of application of **SytheOcc**. We enable geometric-controlled generation that conveys the user editing in 3D voxel space to generate realistic street view images. In this case, we create a rare scene that traffic cones block the way. This advancement facilitates the evaluation of autonomous systems, such as the end-to-end planner VAD (Jiang et al., 2023), in simulated corner case scenes.

applications, such as occupancy prediction and editing scene geometry to create long-tailed scenes, where granular volumetric control is paramount in both tasks.

ControlNet (Zhang et al., 2023) and GLIGEN (Li et al., 2023c) is another type of prominent method in the field of controllable image generation. These approaches exhibit several desirable attributes in terms of controllability. They leverage conditional images such as semantic masks for control, thereby offering a unified framework to manipulate both foreground and background. However, despite its precise spatial control, ControlNet does not align with our specific requirements. Their conditions of pixel-level images differ fundamentally from what we require in 3D contexts. Our experimental results also find that ControlNet struggles to handle overlapping objects with varying depths (see Fig. 6 (a)), as it only utilizes an ambiguous 2D semantic map as conditional input. As a result, it is non-trivial to extend the ControlNet framework and convey their desirable attributes for 3D conditioning.

To address the above challenges, we propose an innovative representation, 3D semantic multi-plane images (MPIs), which contribute to image generation with finer geometric control. In detail, we employ multi-plane images (Zhou et al., 2018) to represent the occupancy, where each plane represents a slice of semantic label at a specific depth. Our 3D semantic MPIs not only preserve accurate and authentic 3D information, but also keep pixel-wise alignment with the generated images. We additionally introduce the MPI encoder to encode features, and the reweighing methods to ease the training with long-tailed cases. As a collection, our framework enables 3D geometry and semantic control for image generation and further facilitates corner case evaluation as depicted in Fig. 1. Finally, experimental results demonstrate that our synthetic data achieve better recognizability, and are effective in improving the perception model on occupancy prediction. In summary, our contributions include:

- We present **SytheOcc**, a novel image generation framework to attain finer and precise 3D geometric control, thereby unlocking a spectrum of applications such as 3D editing, dataset generation, and long-tailed scene generation.

- Incorporating the proposed 3D semantic MPI, MPI encoder, and reweighing strategy, we deliver a substantial advancement in image quality and recognizability over prior works.

- Our extensive experimental results demonstrate that our synthetic data yields an effective data augmentation in the realm of 3D occupancy prediction.

## 2 RELATED WORK

### 2.1 3D OCCUPANCY PREDICTION

The task of 3D occupancy prediction aims to predict the occupancy status of each voxel in 3D space, as well as its semantic label if occupied. Compared with previous perception methods like

3D object detection, occupancy prediction offers a more detailed and nuanced understanding of the environment, as it provides finer geometric details, is capable of handling general, out-of-vocabulary objects, and finally, enriches the planning stack with comprehensive 3D information. Early methods exploited LiDAR as inputs to complete the 3D occupancy of the entire 3D scene (Yan et al., 2021; Mei et al., 2023). Recent methods began to explore the more challenging vision-based 3D occupancy prediction (Wang et al., 2023b; Tian et al., 2024; Tong et al., 2023; Wei et al., 2023). By predicting the geometric and semantic properties of both dynamic and static elements, 3D occupancy prediction offers a more comprehensive understanding of the surrounding environment.

## 2.2 DIFFUSION-BASED IMAGE GENERATION

Recent advancements in diffusion models (DMs) have achieved remarkable progress in image generation. In particular, Stable Diffusion (SD) (Rombach et al., 2022) employs DMs within the latent space of autoencoders, striking a balance between computational efficiency and high image quality. Beyond text control, there is also the introduction of additional control signals. A noteworthy work is ControlNet (Zhang et al., 2023), which incorporates a trainable copy of the SD encoder to extract the feature of conditional images and adds it to the UNet feature. It significantly enhances the controllability and unlocking pathways for advanced applications. We refer readers to recent survey (Yang et al., 2023b) for more details.

## 2.3 IMAGE GENERATION IN AUTONOMOUS DRIVING

As training neural networks relies heavily on labeled data, numerous studies are delving into dataset generation to boost training. Lift3D (Li et al., 2023a) designs generative NeRF to synthesize labeled datasets for 3D detection for the first time. Several other works employ BEV layouts to synthesize image data, proving beneficial for perception models. For example, BEVGen (Swerdlow et al., 2024) conditions BEV layouts to generate multi-view street images, while BEVControl (Yang et al., 2023a) separately generates foregrounds and backgrounds from BEV layouts. MagicDrive (Gao et al., 2024) generates images with 3D geometry controls by independently encoding objects and maps through a text encoder or map encoder. Compared with MagicDrive, our geometry control is characterized by a more detailed and lossless representation of 3D scenes for control, which poses significant challenges than projected layout or box embedding.

Recently, DriveDreamer (Wang et al., 2023a), DrivingDiffusion (Li et al., 2023b), Drive-WM (Wang et al., 2023c) and Panacea (Wen et al., 2023) use a ControlNet framework, which involves projecting bounding boxes and road maps onto 2D FoV images as a conditioning input. This approach has proven to be effective for geometric control. However, it is limited in that it only achieves alignment at the 2D-pixel level. Consequently, this method falls short in capturing the depth hierarchy and fails to account for the occlusion relationships present in the 3D real world. Besides, adding a depth channel like Panacea (Wen et al., 2023) may address the limitations of depth order, but it discards the occluded part and only contains partial observation. UrbanGiraffe (Yang et al., 2023d) train a generative NeRF to perform image generation. WoVoGen (Lu et al., 2023) creates a 4D world volume feature using occupancy to guide the generation, but seems to rely on object mask guidance.

As described above, most of the prior work is restricted by only modeling a projected primitive of 3D boxes and road maps as conditions. They suffer from ill-posed un-projection ambiguity. In contrast, we model 3D occupancy labels as conditions, as they provide finer geometric details and semantic information. However, designing an input representation of 3D occupancy labels into a 2D diffusion model is challenging. In this paper, we propose a novel representation: 3D semantic Multi-Plane Images (MPIs) as conditional inputs, which not only provide spatial alignment that improves visual consistency, but also encode comprehensive 3D geometric information including occluded parts.

## 3 METHOD

**Overview**    The overview of our method is depicted in Fig. 2. Built upon the SD pipeline, we aim to perform geometry-controlled image generation by conditioning on 3D geometry labels with semantics (occupancy labels). One requirement is that the images should faithfully align with the given label. This task is more challenging than conditioned on 3D box due to the sparse and irregular nature of occupancy. We first discuss how to efficiently represent occupancy in Sec. 3.2, followed

Figure 2: The overall architecture of **SytheOcc**. We achieve 3D geometric control in image generation by utilizing our proposed 3D semantic multiplane images to encode scene occupancy. In our framework, we can edit the occupied state and semantics of every voxel in 3D space to control the image generation, thereby opening up a wide spectrum of applications as shown in the top right.

by our designed MPI encoder to enhance generation quality in Sec. 3.3, and reweighing strategy to handle the long-tailed depth and category in Sec. 3.5.

## 3.1 Representation of Condition: Local Control Aligns Better than Global Control

One of the key challenges is how to represent our conditional occupancy input. A straightforward method (Gao et al., 2024; Chen et al., 2023) is to convert the 3D occupancy voxel to 1D global embedding that is similar to text embedding, and then use cross-attention to learn controllable generation. However, these global methods can be less effective when dealing with dense or irregular data due to the following reasons: **(i)** They perform controllable generation through hard encoding the spatial relationship between 1D global embedding and 2D UNet features. **(ii)** Ignore the underlying geometry alignment between the conditional input and the generated image. In contrast, local methods like ControlNet, directly add spatial features to the UNet features, providing 2D local control with pixel-level spatial alignment. They are better than the global method (see Tab. 1), but suffer from 3D ambiguity (see Fig. 6 (a)). Consequently, this comparison motivates us to seek a more compact and efficient manner to encode and condition our 3D occupancy labels.

## 3.2 Represent Occupancy as 3D Semantic Multiplane Images

It is non-trivial to design a 3D representation for conditioning. To efficiently store both the semantic and geometric information of the irregular occupancy input, we propose to use multiplane images (MPIs) (Zhou et al., 2018) as representation. An MPI is composed of a series of fronto-parallel RGBA layers within the frustum of the source camera with a specific viewpoint. These planes are arranged at varying depths, from $d_{min}$ to $d_{max}$, starting from the nearest to the farthest. Each layer of these images contains both an RGB image and an alpha map, which collectively capture the visual and geometric details of the scene at the respective depth. In our work, instead of storing RGB value and alpha map in the original MPI, we store our 3D semantic labels. Each layer of MPI represents the semantic index at the corresponding depth. We display the colored MPI in the top row of Fig. 2 for visual clarity, but we actually use the integer index for learning. We obtain our 3D semantic MPI by:

$$P_l = (u \times d_l, \ v \times d_l, \ d_l)^T, \ d_l = d_{min} + (d_{max} - d_{min}) \times l/D, \tag{1}$$

$$\text{MPI}_{n,l} = \text{Interpolate}(\text{Occupancy}, \ \mathbf{T_n} \cdot \mathbf{K_n^{-1}} \cdot P_l), \tag{2}$$

$$\text{MPI} = \text{Concatenate}(\text{MPI}_{i,j}), \ i \in (0, N), \ j \in (0, D), \tag{3}$$

where $(u, v)$ is a pixel coordinate in image space, $d_l$ is depth value of the $l^{th}$ layer, $n$ denotes the $n^{th}$ camera view. This equation implies we first back project points $P$ in camera frustum space $(u, v, d)$ to Euclid space $(x, y, z)$ by multiplying inverse intrinsic $\mathbf{K^{-1}}$. Then we use transformation matrix $\mathbf{T}$ to map points from camera coordinates to occupancy coordinates. We then use the point coordinates

Figure 3: Visualizations of geometric controlled generation. **Top row**: Fusion of 3D semantic MPI. **Bottom row**: our generation concatenated from neighboring views.

to interpolate the nearest semantic index from the dense occupancy voxel to form a slice of MPI. Finally, we concatenate all slices to form $\texttt{MPI} \in \mathbb{R}^{N \times D \times H \times W}$, where $D$ is the number of layers that is set at 256, $N$ is the number of camera views in the case of batch size = 1.

By representing occupancy as 3D semantic MPI, every pixel in MPI contains geometry and semantic information with implicit depth, seamlessly integrating occluded elements, and ensuring a precise spatial alignment with the generated images.

### 3.3 3D SEMANTIC MPI ENCODER

To enable local control with spatially aligned conditions, we develop a simple but effective MPI encoder that aligns the 3D multi-plane feature to the latent space of the diffusion model. The purpose of the MPI encoder is to obtain features from multi-plane images to perform 3D-aware image synthesis. Unlike the original ControlNet which downsampling conditional input through 3×3 convolutions with padding, we design a 1×1 convolutional encoder without downsampling to encode features. In detail, the 3D multiplane features which have the sample resolution with latent features, are transformed by a 1×1 convolution layer and ReLU activation (Agarap, 2018) in the MPI encoder.

After obtaining the multi-scale feature after the MPI encoder, we add the feature to the decoder of diffusion UNet to provide spatial features. Experimental results in Tab. 2 will show that our 1×1 conv in MPI encoder is more effective than 3×3 conv, as the 1×1 conv with receptive field = 1 provides a spatial align feature to the latent feature in the diffusion UNet. In contrast, 3×3 conv is conducted in a camera frustum space rather than Euclid space, making an imprecise correspondence between 3D multiplane features and 2D image features. Moreover, using 3×3 conv to process 3D semantic MPI will introduce a large computational burden as the channel number increases from 3 channels of RGB to 256 planes. We display our 3D geometry and semantic control property in Fig. 3.

In summary, we chose MPIs as the representation because they **(i)** Incorporate lossless 3D information, including scene geometry rather than 2.5D depth. **(ii)** Provide spatially aligned conditional features that naturally extend the ControlNet framework from image level to 3D level. **(iii)** Capable of representing geometry and semantics including occluded elements.

### 3.4 CROSS-VIEW AND CROSS-FRAME ATTENTION

The sensor arrangement in a self-driving car usually requires a full surround view of cameras to capture the entire 360-degree environment. To effectively simulate the multi-view and subsequent multi-frame generation, zero-initialized (Zhang et al., 2023) cross-view and cross-frame attention are integrated into the diffusion model to maintain consistency between views and frames. Following prior work (Wu et al., 2023; Gao et al., 2024; Wen et al., 2023; Wang et al., 2023c), each cross-view attention allows the target view to access information from its neighboring left and right views, thus training cross-view attention using multi-view consistent images will enforce it to generate the same instance in the overlapping region of multi-view cameras.

$$\texttt{Attention}(Q, K, V) = \texttt{softmax}(\frac{QK^T}{\sqrt{d}}) \cdot V, \tag{4}$$

$$h_{out} = h_{in} + \sum_{i \in \{l,r\}} \texttt{Attention}(Q_{in}, K_i, V_i), \tag{5}$$

where $l$, and $r$ is the camera view of left and right. $Q_{in}$ and $h_{in}$ denotes the query and the hidden state of input view. Similarly, we add cross-frame attention that attend previous frame and future frame to enable video generation. In this case, we use the same formulation while $i \in \{f, h\}$, where $f$ and $h$ is the camera view of future and history frames.

### 3.5 Importance Reweighing

To deal with the extreme imbalance problem between foreground, background, and object categories, and also to ease the training, we propose three types of reweighting methods to improve the generation quality of foreground objects.

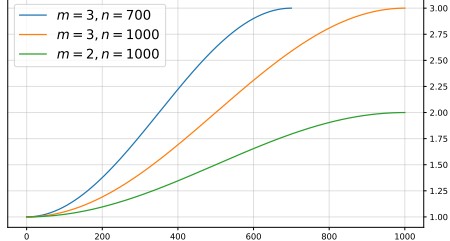

**Progressive Foreground Enhancement**   To mitigate the complexity of the learning task, we propose a progressive reweighting method that incrementally enhances the loss associated with the foreground regions (based on semantic class) as the training progresses. The detailed formulation is:

$$w(x, m, n) = \frac{(m-1)}{2} \cdot (1 + \cos(\frac{x}{n} \cdot \pi + \pi)) + 1, \quad (6)$$

Figure 4: Visualizations of the reweighing function in Eq. 6.

where $x$ is the current training step, $m$ is the maximum value of weights that set at 2, and $n$ is the total training steps. This approach is engineered to facilitate a learning trajectory that progresses from simplicity to complexity, thereby aiding in the convergence of the model. This curve can be interpreted as a cosine annealing but inverted to amplify the importance of the foreground region.

**Depth-aware Foreground Reweighing**   In the meantime, we acknowledge the learning difficulty in different depth places in 3D scenes. Following GeoDiffusion (Chen et al., 2023), we perform depth reweighing to foreground objects by adaptively assigning higher weights to farther foreground areas. This enables the model to focus more thoroughly on hard examples with depth-aware importance reweighting. Instead of using their exponential function to increase weights, we use our designed cosine function Eq. 6 for stability. Here $x$ is the input depth value, and $n$ is the maximum depth that set at 50.

**CBGS Sampling**   To deal with the class imbalance problem in driving scenarios, where certain object categories appear infrequently, we employ the Class-Balanced Grouping and Sampling (CBGS) (Zhu et al., 2019) to better handle the long-tailed classes. CBGS addresses the challenge of class imbalance by grouping and re-sampling training data to ensure each group has a balanced distribution of sample frequency across different object categories. This method reduces the bias towards more frequent classes and enables better generalization to rare scenarios.

### 3.6 Model Training

To ease the training of the MPI encoder and added attention module, we use a two stage training pipeline. We first train MPI encoder and cross-view attention in a multi-view image generation setting. Then we train cross-frame attention and freeze other components in a video generation setting.

**Objective Function**   Our final objective function can be formulated as a standard denoising objective with reweighing:

$$\mathcal{L} = \mathbb{E}_{\mathcal{E}(x), \epsilon, t} \| \epsilon - \epsilon_\theta(z_t, t, \tau_\theta(y)) \|^2 \odot w, \quad (7)$$

where $w$ is the multiplication of progressive reweighing and depth-aware reweighing.

## 4 Experiments

### 4.1 Dataset and Setups

We conduct our experiments on the nuScenes dataset (Caesar et al., 2020), which is collected using 6 surrounded-view cameras that cover the full 360° field of view around the ego-vehicle. It contains 700 scenes for training and 150 scenes for validation. We resize the original image from 1600 × 900 to 800 × 448 for training. In our work, we use the occupancy label with a resolution of $0.2m$ from OpenOccupancy (Wang et al., 2023b) as condition input, while the benchmark of occupancy prediction uses a resolution of $0.4m$ from Occ3D (Tian et al., 2024) dataset for its popularity.

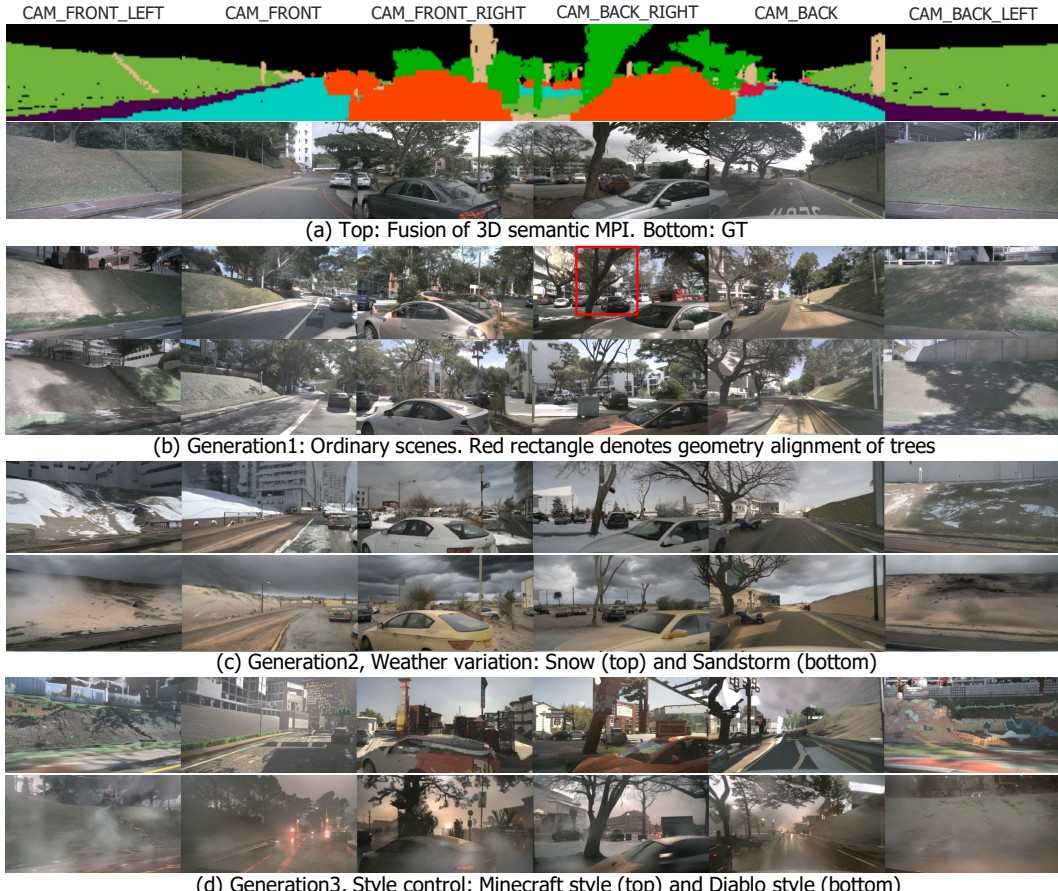

(a) Top: Fusion of 3D semantic MPI. Bottom: GT

(b) Generation1: Ordinary scenes. Red rectangle denotes geometry alignment of trees

(c) Generation2, Weather variation: Snow (top) and Sandstorm (bottom)

(d) Generation3, Style control: Minecraft style (top) and Diablo style (bottom)

Figure 5: Visualizations of generated multi-view images. The generation conditions (occupancy labels) are from nuScenes validation set. We highlight that **(i)** Geometry alignment of trees in red rectangle in (b). **(ii)** Use text prompt to control high-level appearance in (c,d).

**Networks** We use Stable Diffusion (Rombach et al., 2022) v2.1 checkpoint as initialization and only train occupancy encoder, cross-view attention. We additionally add cross-frame attention if in video experiments. We adopt FB-Occ (Li et al., 2023d) as the target model for occupancy prediction for its SOTA performance in this task. The pretrained checkpoint of the network is obtained from their official repository. Since FB-Occ predicts occupancy using only single frame images, we thus train SyntheOcc without cross-frame attention in related experiments. For video generation, we provide experimental results in appendix.

**Metrics** We use Frechet Inception Distance (FID) (Heusel et al., 2017) to measure the perceptual quality of generated images, and use mIoU to measure the precision of occupancy prediction.

**Hyperparameters** We set $D = 256$, $d_{min} = 0$ and $d_{max} = 50$. The depth resolution of MPI is thus higher than occupancy voxel. We train our model in 6 epochs with batch size $= 8$. The learning rate is set at $2e^{-5}$. The training phase takes around 1 day using 8 NVIDIA A100 80G GPUs. We use UniPC scheduler (Zhao et al., 2023) with the classifier-free guidance (CFG) (Ho & Salimans, 2022) that is set as 7.0. During inference, we use 20 denoising steps for dataset generation.

**Baselines** We compare our method with prior methods in Tab. 1. ControlNet denotes we train a ControlNet using an RGB semantic mask as the condition. ControlNet+depth denotes we add a depth channel after the semantic mask to provide 2.5D depth information. The depth map rendered by occupancy is normalized to [0-255] to accommodate the RGB value. The ControlNet+depth can be regarded as a degradation of SytheOcc which is reduced to a single plane. Then we evaluate MagicDrive since it is the only open-sourced method in this area. MagicDrive separately encodes foreground and background using prompt and BEV layout. Furthermore, we evaluate the image

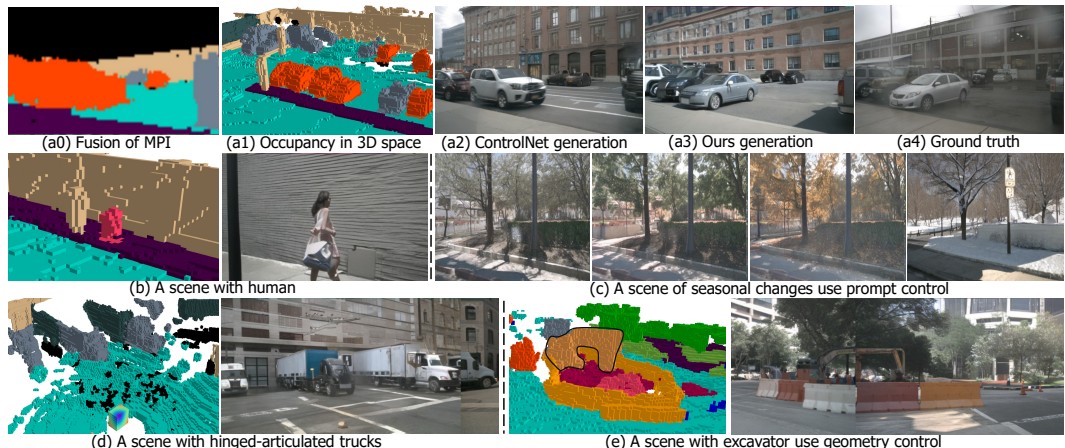

Figure 6: **Top row**: Comparison with ControlNet. We achieve a precise alignment between conditional labels and synthesized images, while ControlNet generates objects with incorrect pose due to ambiguous 2D condition. **Mid and Bottom row**: Visualizations of geometry-controlled image generation. We can faithfully generate objects with the desired topology in a specific 3D position.

quality (FID (Heusel et al., 2017)) of our method in Tab. 3. Compared with prior methods, we use a unified 3D representation that seamlessly handles foreground and background, surpassing them by a large margin.

## 4.2 QUALITATIVE RESULTS

**High-level Control using Prompt**  In Fig. 5 (c,d) and Fig. 6 (c), we demonstrate the capability to employ user-defined prompts to generate images with specific weather conditions and high-level style. Although the nuScenes dataset doesn't contain rare weather images like snow and sandstorms, our method successfully conveys prior knowledge pretrained from stable diffusion to our scenes. Compared with visualization results in prior work like Fig. 8 of MagicDrive, our method shows better alignment with the text prompt, demonstrating the cross-domain generalization ability of our method.

**3D Geometric Control**  Our flexible framework enables us to create novel scenes by manipulating voxels as displayed in Fig. 1 and Fig. 3. Basically, we can edit the occupied state and semantics of every voxel in our scenes for generation. We highlight that we can create a hinged-articulated truck and an excavator as shown in Fig. 6 (d,e). The generated excavator image exhibits a remarkable alignment with the input occupancy that is delineated by a black outline.

**Long-tailed Scene Generation**  The flexibility of 3D semantic MPI has conferred significant advantages upon our approach. In the following, we create long-tail scenes that rarely occur in our real world for evaluation. In Fig. 1, we show that we manually add parallel traffic cones in front of the ego vehicle. This scene has never happened in the training dataset, but our geometric controllability provides us the capability to create such data. We then use the created scene to test autonomous driving systems such as end-to-end planner VAD (Jiang et al., 2023) to validate its effectiveness. In this case, VAD successfully predicts correct waypoints with the high-level command 'turn left'. Moreover, in appendix Sec. B, we generate long-tailed scenes with extreme weather such as snow and sandstorms, and evaluate perception model on it to examine its generalizability of rare weather.

**Comparison with Baselines**  In Fig. 6 (a), we visualize a comparison with ControlNet. We find that ControlNet struggles to distinguish the overlapping instances in 2D-pixel space. This leads to the two parked cars being merged into a single car with incorrect pose. In contrast, our 3D semantic MPIs contain more than 2D semantic mask, but also account for complete scene geometry with occluded parts. Together with our proposed MPI encoder and reweighing strategy, our framework yields a realistic image generation with high-quality label alignment. More comparison is provided in Sec. D.

| Method | Train | Val | mIoU | barrier | bicycle | bus | car | cons. veh. | moto. | pedes. | traf. cone | trailer | truck | drive. suf. | other flat | sidewalk | terrain | manmade | vegetation |
|---|---|---|---|---|---|---|---|---|---|---|---|---|---|---|---|---|---|---|---|
| Oracle (FB-Occ) | Real | Real | 39.3 | 45.4 | 28.2 | 44.1 | 49.4 | 25.9 | 28.8 | 28.0 | 27.7 | 32.4 | 37.3 | 80.4 | 42.2 | 49.9 | 55.2 | 42.0 | 37.7 |
| **SytheOcc**-Aug | Real+Gen | Real | 40.3 | 45.4 | 27.2 | 46.6 | 49.5 | 26.4 | 27.8 | 28.4 | 29.4 | 34.0 | 37.2 | 81.3 | 46.0 | 52.4 | 56.5 | 43.3 | 38.9 |
| MagicDrive | Real | Gen | 13.4 | 0.7 | 0.0 | 11.8 | 32.4 | 0.0 | 6.6 | 2.8 | 0.3 | 2.6 | 19.6 | 60.1 | 12.1 | 26.2 | 23.4 | 15.5 | 12.8 |
| ControlNet | Real | Gen | 17.3 | 17.7 | 0.2 | 13.6 | 21.0 | 0.6 | 0.8 | 8.6 | 10.4 | 6.9 | 11.9 | 67.4 | 18.8 | 36.4 | 36.9 | 20.8 | 22.4 |
| ControlNet+depth | Real | Gen | 17.5 | 19.3 | 0.3 | 14.0 | 23.7 | 1.0 | 0.6 | 9.2 | 9.2 | 5.7 | 12.1 | 68.8 | 19.2 | 36.0 | 35.3 | 19.8 | 22.8 |
| **SytheOcc**-Gen | Real | Gen | **25.5** | 32.6 | 13.8 | 27.7 | 33.4 | 7.5 | 6.5 | 15.7 | 16.5 | 16.5 | 25.6 | 74.3 | 24.5 | 39.4 | 40.5 | 28.6 | 28.8 |

Table 1: Downstream evaluation on the **nuScenes-Occupancy** validation set. Based on the used train and val data, two types of settings are reported. The first is to use generated training set to augment the real training set, and evaluate on the real validation set, denoted as Aug. The second is to use pretrained models trained on the real training datasets to test on the generated validation set, denoted as Gen.

## 4.3 QUANTITATIVE RESULTS

**Recognizability, Realism and Controllability Evaluation**    To evaluate whether our generated images aligned with given annotations, we provide Gen experiment in Tab. 1. Using the annotation of val set, we synthesize a copy of val set's images, then use perception model trained on real training set to perform evaluation. The performance will be more effective as it is close to the oracle performance. We find that local method (ControlNet) perform better than global method (MagicDrive). Furthermore, SytheOcc generalizes the locality for 3D conditioning to yield better performance.

**Data Augmentation for 3D Occupancy Prediction**    Notably, we conduct experiments using our synthesized dataset to enhance the real training set in Tab. 1. We first use the occupancy labels from training set to create a synthetic training set. Then we modify the loading pipeline in perception model to randomly sample images from real dataset or synthetic dataset and train network from scratch. Therefore, our approach preserves the inherent training dynamics of the neural network by solely modifying the training images, without any alteration to the number of training iterations or epochs. As MagicDrive-Aug exhibits numerical overflow when training FB-Occ, which may attributed to unsatisfactory recognizability, we have to omit it and only provide MagicDrive-Gen experiments. The experiment of ControlNet(+depth)-aug is provided in appendix Sec. E.

As shown in Tab. 1, where SytheOcc-Aug denotes the augmentation experiments using our generated dataset, shows a satisfactory improvement over the prior state of the art. We emphasize that surpassing the performance of the original dataset is not the primary objective of our work; rather, it is an ancillary benefit that emerges from our framework for geometry-controlled generation.

**Ablations**    In Tab. 2, we present ablation studies across several design spaces of our model, analogous to the Gen experiment in Tab. 1. We find that our designed MPI encoder of 1×1 conv has significant improvement compared to the conventional 3×3 conv approach. Besides, our proposed three types of reweighing methods demonstrate a consistent improvement over the baseline. As a result, the improved image quality and label alignment enable higher precision in downstream tasks.

| MPI Encoder | Reweighing Method | | | Metric |
|---|---|---|---|---|
| Design | Progressive | Depth | CBGS | mIoU |
| 3×3 | - | - | - | 21.96 |
| 1×1 | - | - | - | 23.05 |
| 1×1 | ✓ | - | - | 23.63 |
| 1×1 | ✓ | ✓ | - | 24.40 |
| 1×1 | ✓ | ✓ | ✓ | 25.50 |

Table 2: Ablation of different designs of the MPI encoder and reweighing methods.

| Method | Condition Type | FID |
|---|---|---|
| BEVGen (Swerdlow et al., 2024) | BEV map | 25.54 |
| BEVControl (Yang et al., 2023a) | BEV map | 24.85 |
| DriveDreamer (Wang et al., 2023a) | Box + FoV map | 52.60 |
| MagicDrive (Gao et al., 2024) | Box + BEV map | 16.20 |
| Panacea (Wen et al., 2023) | Box + FoV map | 16.96 |
| SyntheOcc | 3D Semantic MPI | **14.75** |

Table 3: Comparison of FID with previous methods on the nuScenes dataset.

## 5 LIMITATION AND BROADER IMPACTS

**Layout Genereation**    Our method is restricted in a conditional generation framework that should have a conditional input at first. Our condition signal is from the original dataset annotation. Thus most of the augmented data is generated using the same occupancy layout, or with minimal human editing. Future research can incorporate the recent research (Lee et al., 2024; Zhang et al., 2024; Lu et al., 2023; Liu et al., 2024; Wu et al., 2024) that generates occupancy and traffic descriptions of the scenes to synthesize images with novel occupancy or traffic layouts.

**Extend to Native Video Diffusion Model**    As our work primarily focus on precise 3D controllability for image generation, we do not specifically tailor for video generation. We have a preliminary attempt to implement a plug-and-play module of cross-view and cross-frame attention to learn view-consistent or frame-consistent generation. This experiment serves as a proof of concept, demonstrating the adaptability of our framework. It shows a promising result of temporal consistency across frames, as compared with prior work in Sec. F. However, we argue that this methodology can be inevitably less significant than native 3D convolution and 3D attention in the video diffusion model. In the future, we plan to extend our controllability into native video diffusion using CogVideoX (Yang et al., 2024) or Open-Sora (Zheng et al., 2024), to further provide controllable video generation.

**Long-tailed Scene Generation**    In this paper, we investigate a series of long-tailed scene generation and corner case evaluations such as rare layout in Fig. 1 and extreme weather in Sec. B. Future work can extend our framework to **(i)** Synthesize more samples for tail classes to boost performance. **(ii)** Generate or replicate large-scale databases of corner cases (Li et al., 2022) for robust perception.

**Closed-loop Simulation**    Given the underlying diverse and controllable image generation of our method, it would be advantageous and valuable to extend our work to a broader domain such as closed-loop simulation (Ljungbergh et al., 2024; Yang et al., 2023e), to enable high-fidelity autonomous systems testing. This line of work can be conducted by utilizing motion conditions to generate future frames as in world model (Yang et al., 2023c; Wang et al., 2023c; Lu et al., 2023), or by explicitly modeling scene graph as in the case of UniSim (Yang et al., 2023e; Ost et al., 2021) and NeuroNCAP (Ljungbergh et al., 2024).

## 6 CONCLUSION

In this paper, we propose **SytheOcc**, an innovative image generation framework that is empowered with geometry-controlled capabilities using occupancy. We introduce a novel 3D representation, 3D semantic MPIs, to address the critical challenge of how to efficiently encode occupancy. This representation not only preserves the authentic and complete 3D geometry details with semantics, but also provides a spatial-align feature representation for 2D diffusion models. With this property, our method enjoys photorealistic appearances and fine-grained 3D controllability, serves as a generative data engine to enable a broad range of applications. Extensive experiments demonstrate that our synthetic data facilitate the training for perception models on occupancy prediction, and provide valuable corner case evaluation in a simulated world.

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

# APPENDIX

In the appendix, we provide the following content:

| | |
|---|---|
| Sec. A: Potential Discussion. | Sec. F: Results of Video Generation. |
| Sec. B: Long-Tailed Scene Evaluation. | Sec. G: Generalize to New Cameras. |
| Sec. C: Ablation of plane number in MPIs. | Sec. H: Impact of Amount of Augment Data. |
| Sec. D: Additional Qualitative Comparison. | Sec. I: Visualization of Failure Cases. |
| Sec. E: Additional Experiments. | |

## A  POTENTIAL DISCUSSION

To help a comprehensive understanding of our paper, we discuss intuitive questions that might be raised.

**How to define geometric control?**   In our paper, we refer the geometric controllable generation as using a voxel grid in 3D space to control the image generation. Although the voxel is a quantized representation of the 3D world, when the resolution goes larger, it can already faithfully represent the geometry detail of scenes. Currently, we are limited by the precision of ground truth labels. The $0.2m$ occupancy grid is a tensor of 500×500×40 that cover a space in x-axis spanning $[-50m, 50m]$, y-axis spanning $[-50m, 50m]$, z-axis spanning $[-5m, 3m]$. In the future, we plan to explore a higher resolution of geometric control to refine our generation.

**Can 3D semantic MPI extend to other representations beyond occupancy?**   Except for occupancy, several other 3D representations can be expressed by 3D semantic MPI, such as mesh, dense point clouds, and even 3D boxes or HD maps. The underlying mechanism is to cast several slices of multi-plane images at different depths to retrieve geometric information. Our application scope is wide, and we left them for future work. As a result, our 3D semantic MPI can be regarded as a general 3D conditioning representation to benefit a wide spectrum of practical systems. These encompass but are not limited to 3D generation such as text2room (Höllein et al., 2023), RoomDreamer (Song et al., 2023), WonderJourney (Yu et al., 2023), and LucidDreamer (Chung et al., 2023), each of which stands to benefit from the rich geometric context provided by our approach.

**Occupancy is complex. How to edit occupancy for controllable generation?**   We agree that occupancy is more complex than the 3D box, but it provides a more nuanced scene description. To ease the editing, we provide a strategy that disentangles the foreground control and background control in occupancy data. If we want to edit a car's trajectory, we can keep the background occupancy unchanged and select the car's first frame occupancy using the 3D box. During the following frames, we remove foreground occupancy and simply place our foreground target's occupancy in a certain location using trajectory. By doing so, we only add minor steps by using occupancy but provide more precise 3D control, which makes it a favorable choice for conditioning.

## B  LONG-TAILED SCENE EVALUATION

In this section, we explore to use SytheOcc to create long-tailed scenes for downstream evaluation. This also stands for evaluating our model using several corner cases. Similar to the SytheOcc-Gen experiment in Tab. 1, we generate a synthetic validation set but use prompts control to manipulate weather patterns or the intensity of illumination.

As depicted in Fig. 10. We create a variety of weather conditions including sandstorms, snow, foggy, rainy, day night, and day time. The motivation behind the creation of these scenes lies in their extreme rarity compared to the ordinary scenes we have captured. The generation of such data is of significant value, as it aids in addressing the long-tailed distribution of scenes, thereby enriching the diversity of our dataset. More visualization is provided in Fig. 12.

In Tab. 4, we observe that all kinds of extreme weather lead to a degradation in performance. This observation underscores the limitations of the perception model in terms of its generalizability to infrequent weather scenarios. Among them, we find that foggy, rainy, and day night exert the most

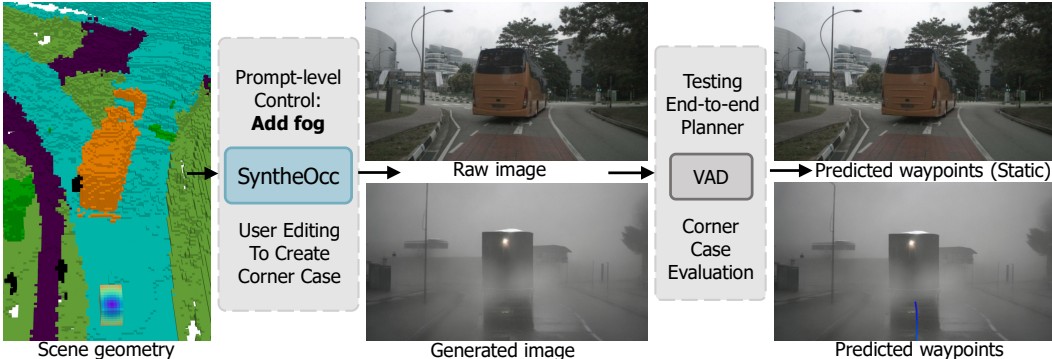

Figure 7: Use **SytheOcc** to create long-tailed scenes for testing. **Top**: In the ordinary scene of a bus placed in front of the ego vehicle, the end-to-end planner VAD (Jiang et al., 2023) predicts future waypoints without movement, thus not plotted in the image. **Bottom**: By harnessing the prompt-level control in our framework, we simulate a scene with the same layout but filled with fog. VAD predicts wrong waypoints that will collide with the bus.

severe impact, as they contribute to a large reduction in visibility as shown in Fig. 10. To improve the generalizability to handle various weather conditions, future work can leverage our generated data to cover the long-tailed scenes, or use adversarial search to find severe scenes based on our framework.

| Scenes | Sandstorm | Snow | Foggy | Rainy | Day night | Day time (raw data) |
|---|---|---|---|---|---|---|
| FB-Occ mIOU | 22.88 | 18.25 | 10.29 | 9.71 | 9.95 | 25.50 |

Table 4: Experiments of downstream evaluation on long-tailed scenes with extreme weather.

Furthermore, we perform long-tailed scene evaluation in Fig. 7. We display the failure of the downstream model VAD (Jiang et al., 2023) in our synthetic long-tailed scene. In this case, we simulate a foggy environment that the dense fog obscures the majority of the ego view. Our experiment reveals that due to the lack of training images of foggy scenes, VAD erroneously predicts waypoints that would result in a collision with the bus. This experiment elucidates the boundaries and failure cases of the VAD model (Jiang et al., 2023). It exposes the limitations of the system under certain conditions, thereby providing insights into scenarios where the model's performance may be compromised.

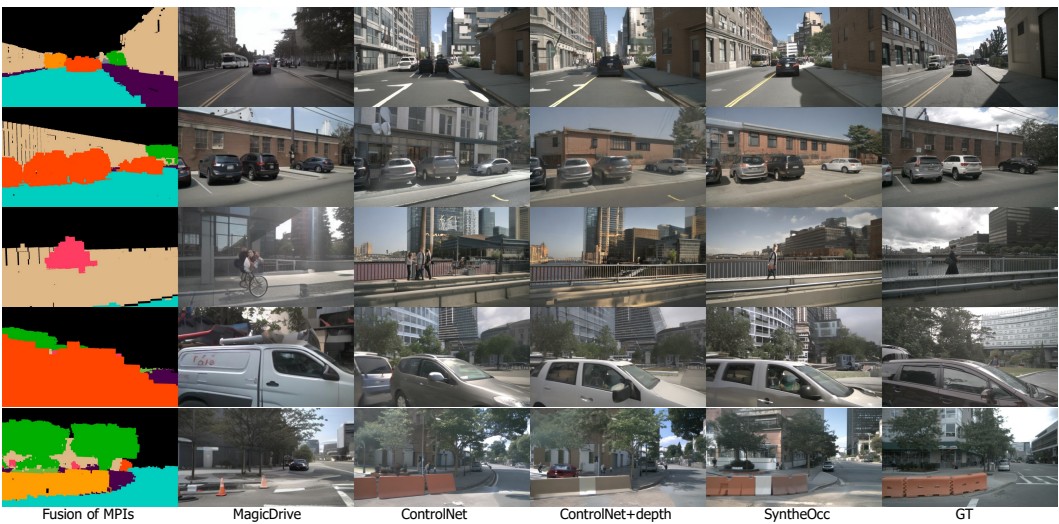

Figure 8: Comparison with baselines.

## C    ABLATION OF PLANE NUMBER OF MPIS

In our proposed 3D semantic MPIs, the number of planes is a hyperparameter that affects the precision of 3D representation. The plane number can be regarded as the 3D resolution in depth axis. The larger the plane number, the MPI will contain more details. We find that an increase in the number of planes is associated with improved accuracy in downstream tasks. This finding denotes that more condition information leads to better downstream task performance.

| Number of Planes | 96 | 128 | 256 |
|---|---|---|---|
| FB-Occ mIOU | 23.36 | 24.28 | 25.50 |

Table 5: Ablation of the number of multi-plane images.

## D    QUALITATIVE COMPARISON WITH BASELINES AND SOTA

In Fig. 8, we conduct a qualitative comparison of our method against MagicDrive, ControlNet, and ControlNet+depth. We find that all the methods display a satisfactory image quality, as they build upon the foundation of the stable diffusion model. The generation of MagicDrive fails to synthesize barriers as shown in the bottom row. ControlNet struggles to generate objects with the correct pose solely from only 2D conditions as shown in the second row. ControlNet+depth, a degradation of our method, an enhancement over ControlNet in terms of alignment, nevertheless suffers from a loss of finer detail in scenes with heavy occlusion, as shown in the human of the third row. Our method, in contrast, aims to address these challenges and provide a more nuanced and accurate generation of complex scenes.

## E    ADDTIONAL EXPERIMENTS

**Data Augmentation using ControlNet**    We provide experiments that use ControlNet and ControlNet+depth to enable data augmentation. This experiment is analogous to the Aug experiment in Tab. 1. In the experiment of ControlNet and ControlNet+depth, due to the potential for input-generation ambiguity, the augmented data could lead to the propagation of inaccurate gradients, thereby affecting the training process. These experiments demonstrate that our approach outperforms the ControlNet baseline in terms of effectiveness.

| Methods | No aug | ControlNet | ControlNet+depth | SyntheOcc |
|---|---|---|---|---|
| **FB-Occ mIOU** | 39.3 | 39.0 | 39.1 | **40.3** |

Table 6: Experiments of evaluating the data augmentation effects using different generative model.

**Evaluate on 3D Detection**    We assess the accuracy of 3D detection using the BEVFusion (Liu et al., 2023). This experiment corresponds to the Generation experiment presented in Tab. 1, with the distinction that BEVFusion is employed for evaluating 3D detection precision. Owing to the effective pixel-aligned alignment offered by our method, SyntheOcc yields superior detection accuracy compared to previous studies.

| Methods | MagicDrive-mAP | MagicDrive-NDS | SyntheOcc-mAP | SyntheOcc-NDS |
|---|---|---|---|---|
| **Results** | 20.8 | 30.2 | **22.3** | **31.3** |

Table 7: Experiments of evaluating the generation quality using 3D detection accuracy.

## F    EXTEND TO VIDEO GENERATION

As described in the main paper Sec. 3.4, we further extend the cross-view attention to cross-frame attention to perform video generation. Our generation results can be find in Fig. 11 and supplementary

video. Our implementation is adopted from MagicDrive (Gao et al., 2024) which is similar to Tune-a-video (Wu et al., 2023). The formulation of cross-frame attention is:

$$\texttt{Attention}(Q, K, V) = \texttt{softmax}(\frac{QK^T}{\sqrt{d}}) \cdot V, \tag{8}$$

$$h_{out} = h_{in} + \sum_{i \in \{f,h\}} \texttt{Attention}(Q_{in}, K_i, V_i), \tag{9}$$

where $f$, and $h$ are the camera view of future and history frames. $Q_{in}$ and $h_{in}$ denotes the query and the hidden state of input view. We train our model in a two-stage pipeline. We first train the MPI encoder and cross-view attention in a multi-view image generation setting. Then we train cross-frame attention and freeze other components in a video generation setting.

In practice, we use the keyframe annotation of the nuScenes dataset to train our video model. We start with our pretrained MPI encoder and cross-view attention and only train our cross-frame attention while keeping others frozen. We employ a sequence of 7 frames as a batch, resulting in a batch size of 42 images for the training process.

We further evaluate the Fréchet Video Distance (FVD) score (Unterthiner et al., 2019) to evaluate the video generation quality in Tab. 8. Attributed to our commendable controllable image generation quality, SyntheOcc achieves competitive performance that is on par with other models.

| Methods | DriveGAN | DriveDreamer | DrivingDiffusion | Ours |
|---|---|---|---|---|
| **FVD** | 502 | 340 | 332 | **251** |

Table 8: Experiments of evaluating the quality of video generation.

Given that our primary contribution does not lie in video generation, this experiment serves as a proof of concept, demonstrating the potential adaptability of our framework. Future research may extend our methodology to facilitate the generation of longer video sequences, thereby expanding the scope and applicability of our framework.

## G  GENERALIZE TO NEW CAMERAS

In this section, we investigate the adaptability of our method to a new set of cameras with different intrinsic. Given that our training set has a fixed camera intrinsic and extrinsic, generalizing to novel cameras indicates that our approach possesses robust generalization capabilities. As shown in Fig. 9, benefiting from our local type of condition, SytheOcc generates images that faithfully align with the new intrinsic, proving that SytheOcc do not over-fit certain parameters. Regarding extrinsic parameters, we can cast our MPI at the desirable locations to retrieve geometric information, thus inherently ensuring generalizability without doubt.

## H  THE INFLUENCE OF THE AMOUNT OF AUGMENTED DATA

As SytheOcc is capable of generating an infinite number of synthetic data, we investigate the influence of the amount of augmented data on downstream tasks in Tab. 9. We find that when our augmented data is expanded from one-fold to two-fold of the training dataset, the performance of perception model slightly decreases. This may indicate the generated data has an optimal ratio for downstream tasks. Due to limited computational resources, we only experiment with a limited amount of ratio. Future work can conduct more thorough experiments to find a universal theorem.

| Amount of Augmented Data | 0 (no augmentation) | 1 | 2 |
|---|---|---|---|
| FB-Occ mIOU | 39.3 | 40.3 | 40.1 |

Table 9: Ablation of the amount of augmented data.

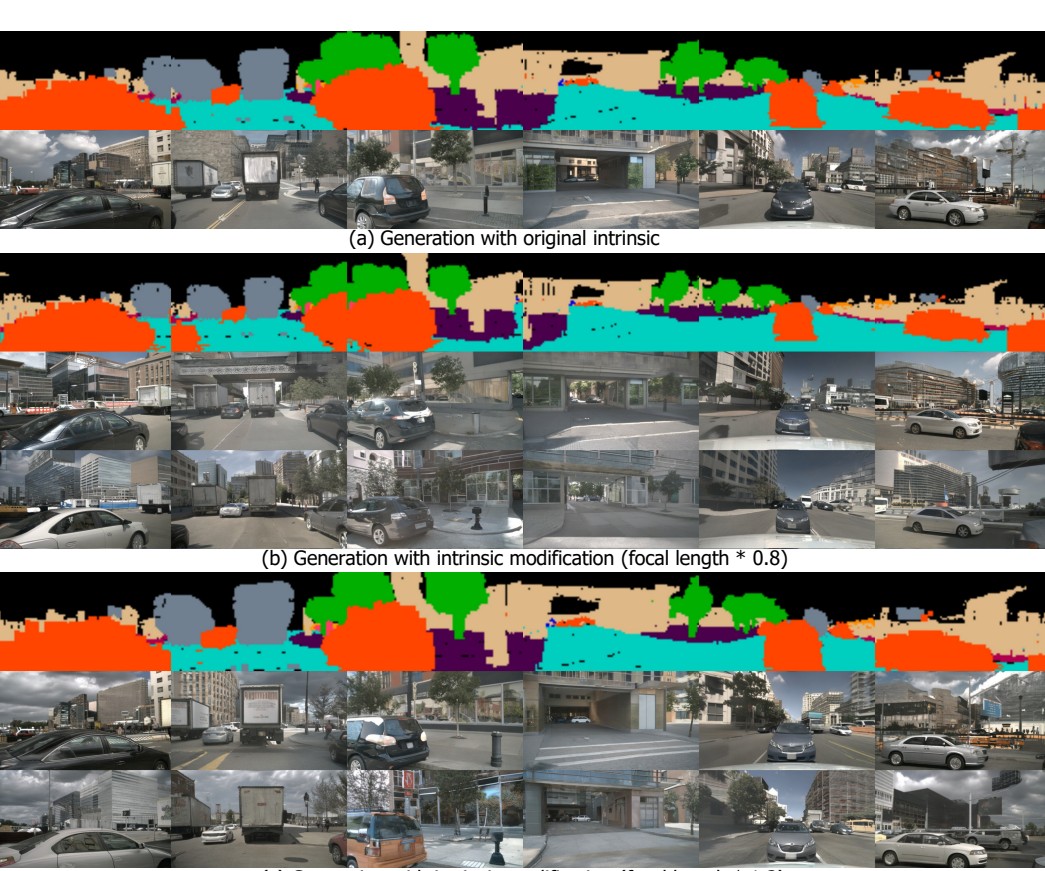

(a) Generation with original intrinsic

(b) Generation with intrinsic modification (focal length * 0.8)

(c) Generation with intrinsic modification (focal length * 1.2)

Figure 9: We demonstrate the generalizability of SytheOcc to new camera intrinsic. We multiply factors to the focal length while keeping the resolution the same. In (b,c), focal length $\times 0.8$ denotes a camera with a larger field of view similar to zoom out, focal length $\times 1.2$ denotes a camera with a smaller field of view similar to zoom in.

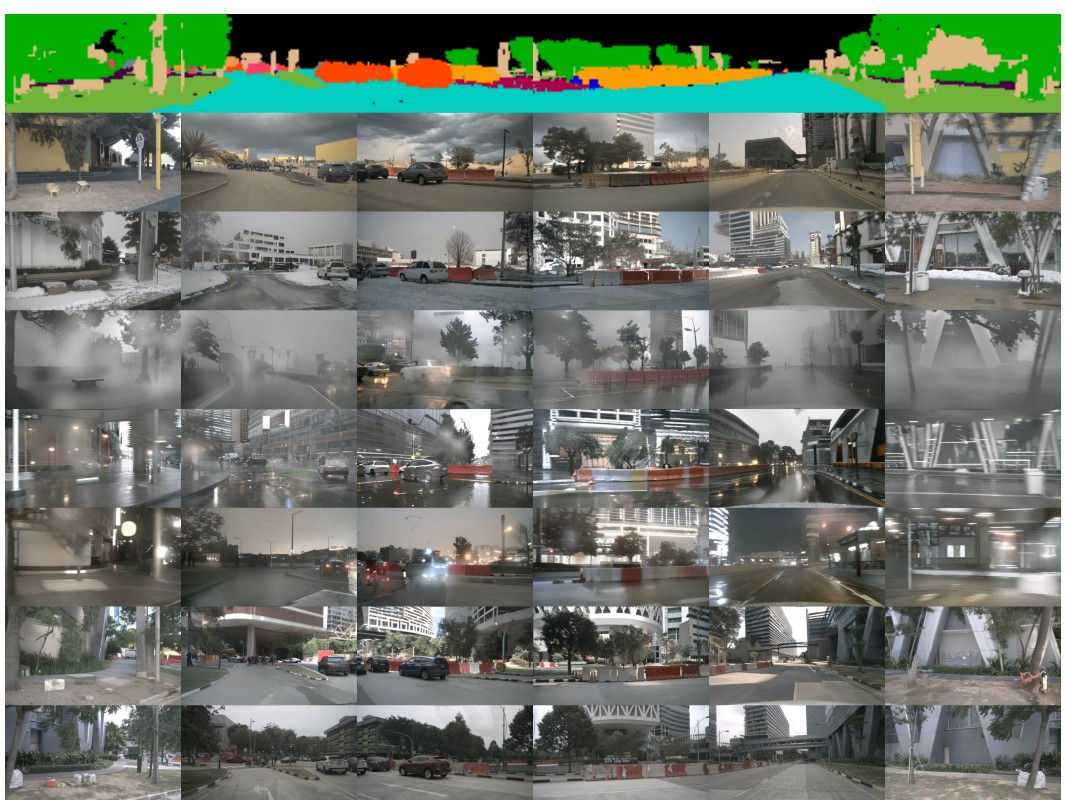

Figure 10: From top to bottom, we display images of fusion of 3D semantic MPI, synthesized images of sandstorm, snow, foggy, rainy, day night, day time, and ground truth.

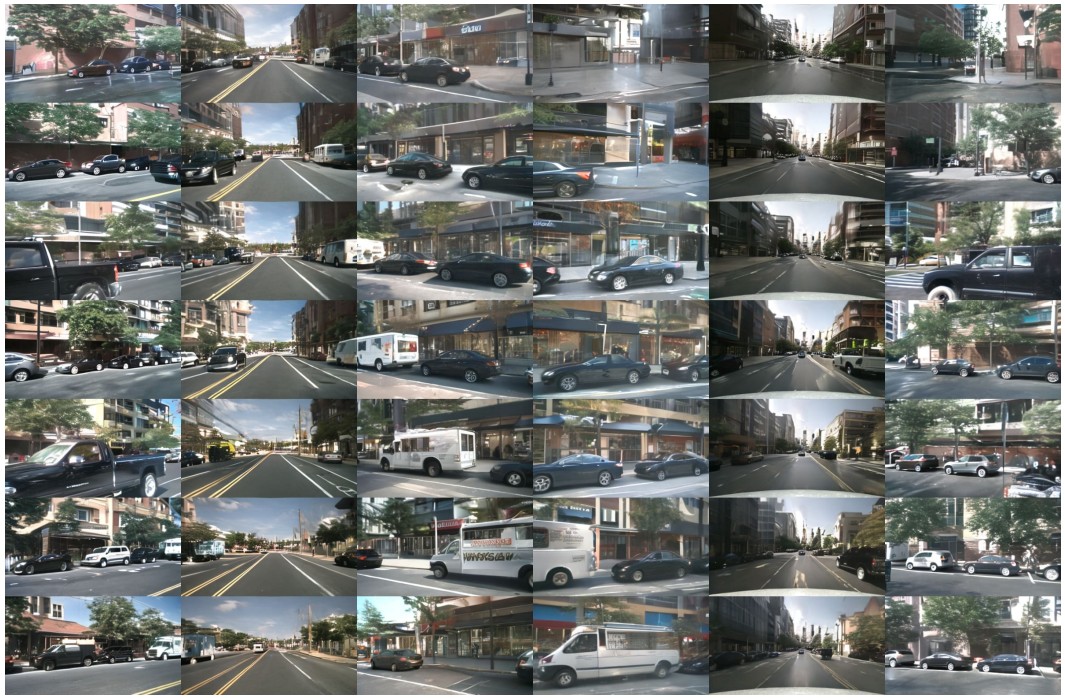

Figure 11: Video generation results of large dynamics scenes. The white car comes across different views and frames depicting consistent shapes with only a slight appearance change.

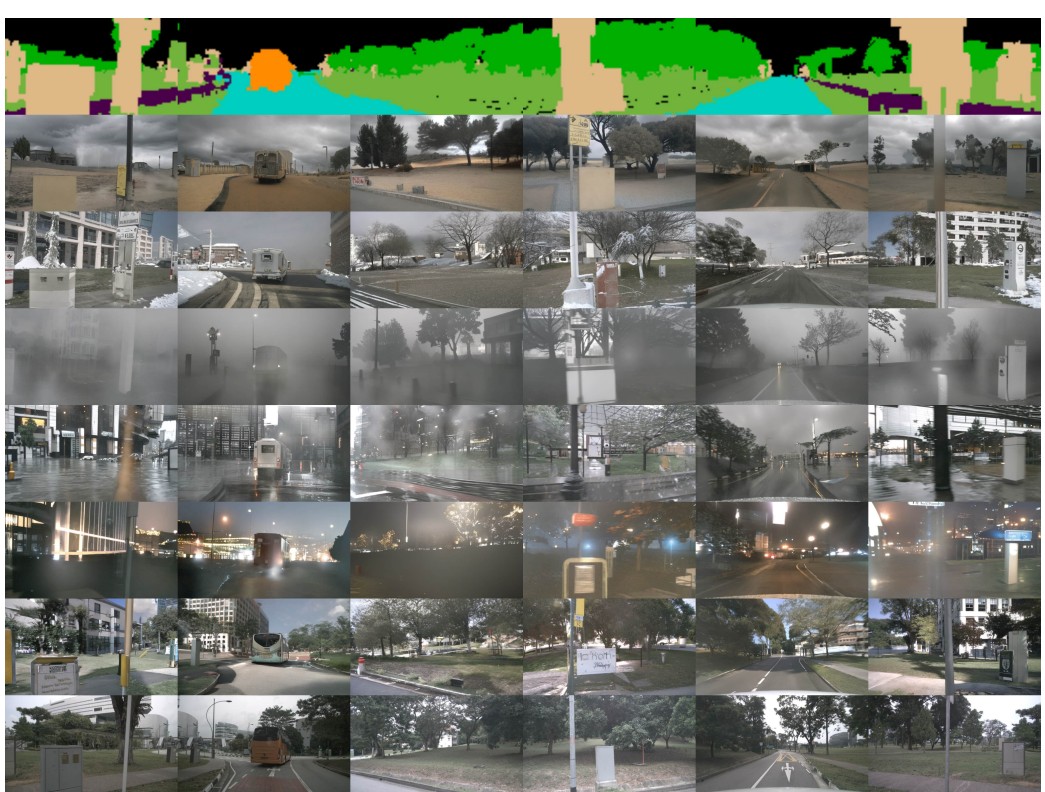

Figure 12: From top to bottom, we display images of fusion of 3D semantic MPI, synthesized images of sandstorm, snow, foggy, rainy, day night, day time, and ground truth.

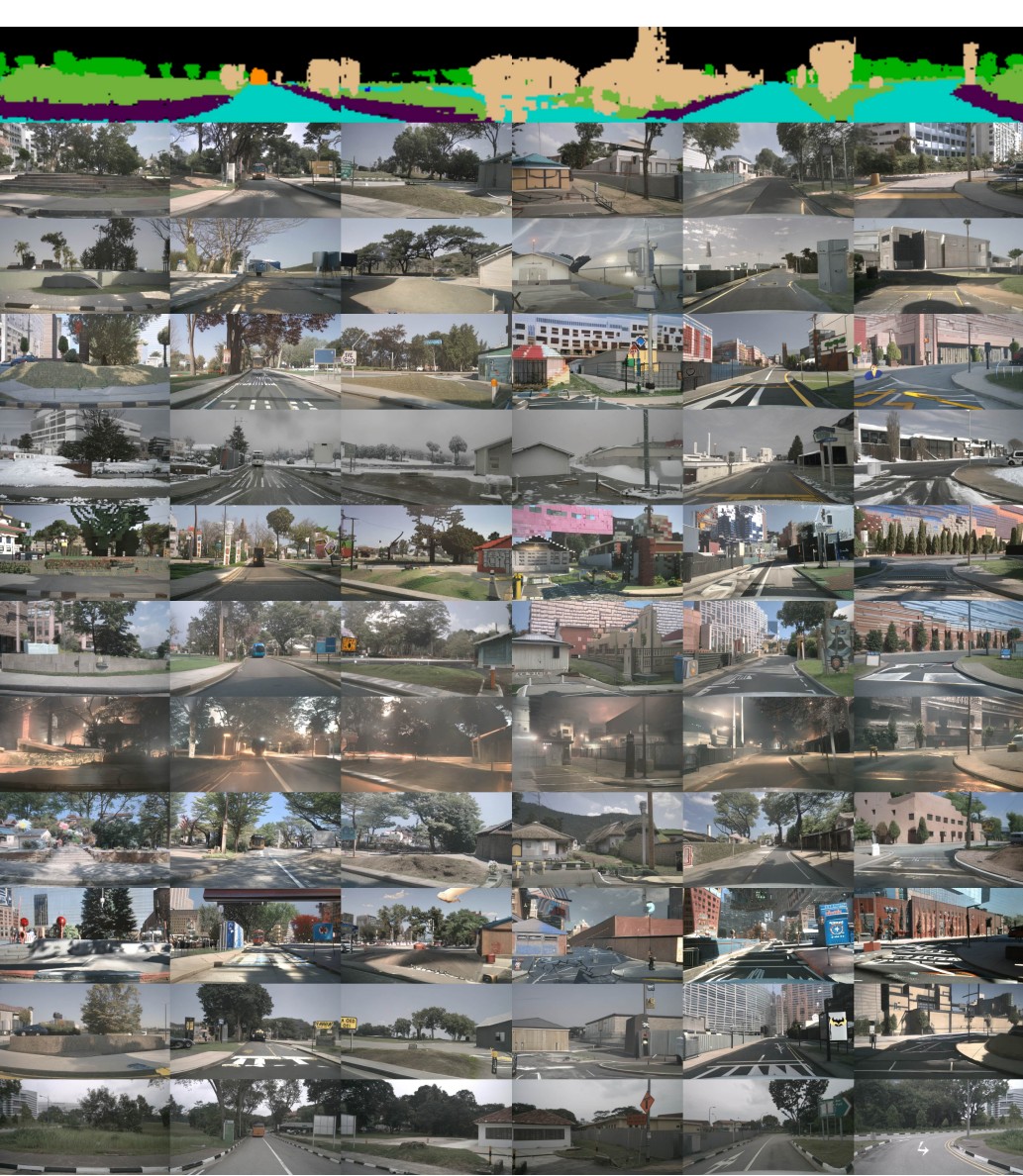

Figure 13: Out of distribution generation. We use prompts to control the high-level appearance of images with specific styles. From top to bottom, we display (1) fusion of 3D semantic MPI. (2) Sunny day. (3) Science fiction style. (4) 8-bit pixel art style. (5) Snowfall. (6) Minecraft style. (7) Pokémon style. (8) Diablo style. (9) Ghibli style. (10) Metropolis style. (11) Gotham style. (12) Ground truth.

## I  FAILURE CASES

We display several failure cases of our method. In Fig. 14, we show a crowd scenes. In this scenario, the excessive number of pedestrians presents a challenge to the cross-view attention and cross-frame attention modules. We find our method incapable of discerning individual entities with clarity. Future research can improve the model capacity or enrich high-quality data to mitigate this problem.

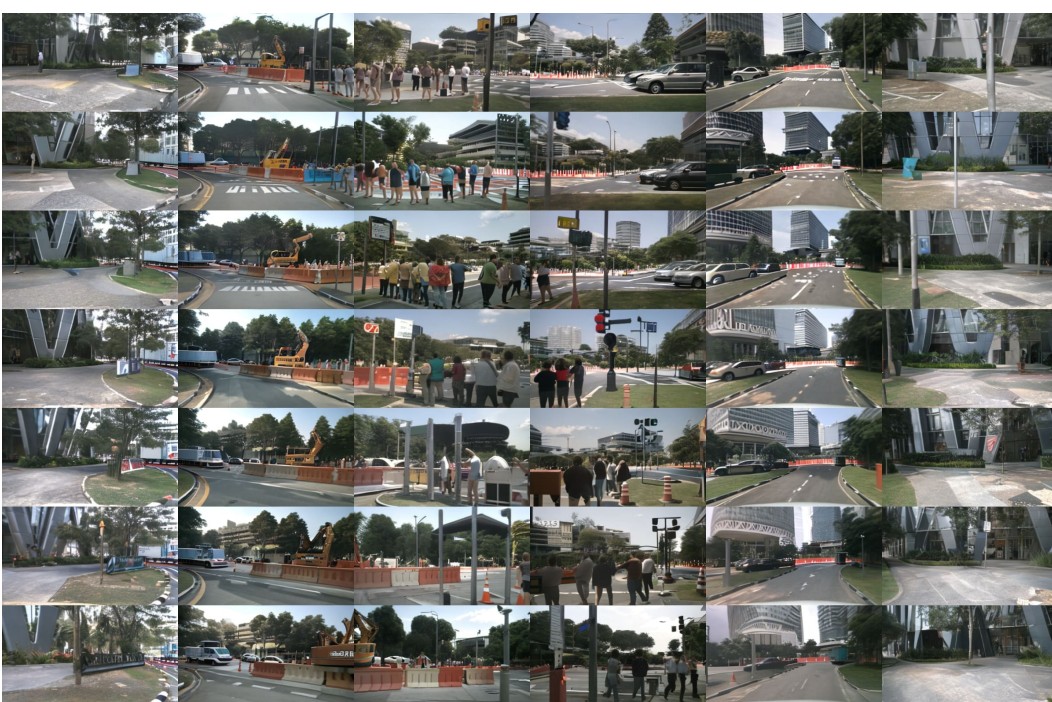

Figure 14: Failure case of video generation results. Our cross-frame attention module is challenging to distinguish a crowd of people across different views and frames.

