# OpenReview forum: "SyntheOcc: Synthesize Geometric-Controlled Street View Images through 3D Semantic MPIs"
_ICLR.cc/2025/Conference — ICLR 2025 Conference Withdrawn Submission_

### Official Review · Reviewer_NHs7 · 2024-10-31

**Soundness:** 2
**Presentation:** 2
**Contribution:** 2
**Rating:** 5
**Confidence:** 5

**Summary:**

The paper proposes SyntheOcc, a novel image generation framework that can synthesize photorealistic and geometric-controlled street view images by conditioning on 3D occupancy labels. The key innovation is the use of 3D semantic multi-plane images (MPIs) to efficiently encode 3D geometric information as conditional input to the 2D diffusion model. The extensive experiments demonstrate that the synthetic data generated by SyntheOcc can effectively augment perception models for 3D occupancy prediction tasks.

**Strengths:**

1. The paper proposes an innovative approach by replacing ControlNet with multiplane images, enhancing image synchronization from occluded views.
2. The provided video effectively demonstrates and supports the proposed method.
3. The writing is clear and easy to follow.
4. The experiments are comprehensive, covering both quantitative and qualitative evaluations.

**Weaknesses:**

1. **Clarification on excluding ControlNet**: The paper should more thoroughly explain the decision to exclude ControlNet. The rationale for why ControlNet fails to meet 3D requirements remains unclear. Since multiplane images could serve as conditions for ControlNet.

2. **Incorporating KPM evaluation for consistency**: It would be beneficial for the paper to include KPM evaluations from Driving into the Future [1] to better assess temporal and multiview consistency.

3. **Additional out-of-domain results**: Presenting more out-of-domain results, such as experiments with variations in camera intrinsic and extrinsic parameters, would explain the ability of generalization of the model.

4. **Weak video quality**: Some objects in the provided video appear twisted or lack realistic representation. And the videos look a little bit unreal but I cannot tell why.

5. **World model integration**: Considering that driving scene generation works nowadays provides world model results—capable of forecasting future layouts and generating future images based on actions. It would be valuable for the paper to explore integration with world models. For example, testing if the generation method can be adapted to synthesize occupancy predictions, as seen in recent work on occupancy-based world models [2]. It will showcase potential for further real-world applications.

[1] Wang, Yuqi, et al. "Driving into the future: Multiview visual forecasting and planning with world model for autonomous driving." CVPR 2024.
[2] Zheng, Wenzhao, et al. "Occworld: Learning a 3d occupancy world model for autonomous driving." ECCV 2024

**Questions:**

I am still wondering why the paper cannot use ControlNet, as multiplane images are still images and could potentially be used as input for ControlNet.

---

### Official Review · Reviewer_KTUH · 2024-10-31

**Soundness:** 3
**Presentation:** 3
**Contribution:** 3
**Rating:** 5
**Confidence:** 4

**Summary:**

This paper introduces SytheOcc, a novel image generation framework enabling precise 3D geometric control for applications like 3D editing and dataset generation. By leveraging 3D semantic multiplane images (MPIs), the framework achieves finer geometry and semantic control, enhancing image quality and recognizability. Experimental results show the effectiveness of synthetic data in augmenting 3D occupancy prediction tasks, indicating a significant advancement over existing methods.

**Strengths:**

1. SytheOcc offers finer and precise 3D geometric control, allowing for intricate manipulation of object shapes and scene geometry, which is crucial for tasks like 3D editing and dataset generation.
2. Experimental results demonstrate that the synthetic data generated by SytheOcc exhibit better recognizability, indicating a substantial advancement in image quality over existing methods.
3. The synthetic data produced by SytheOcc prove to be highly effective for data augmentation in 3D occupancy prediction tasks, enhancing the performance and robustness of perception models in such applications.

**Weaknesses:**

1. The practicality of occupancy editing in 3D space should be addressed. It is crucial to automate or accelerate the editing process to make it feasible for practical applications requiring large amounts of data. The authors may report the time required to generate a new image through editing, and discuss possible solutions to scaling up data.
2. The paper use occupancy as a condition due to its spatial information. It would be beneficial to discuss the fundamental differences between using occupancy and using of depth&segmentation maps as conditions to control image generation [1]. Experimental comparisons between using depth&semantic maps versus occupancy as conditions could be conducted to evaluate metrics like FID and inference time.
3. The multi-view consistency appears not so good. In Figure 5 (b) (c), the color of the car in the first row changes in the second and third images. The authors could includsa comparative analysis of the generation results from different models (e.g., MagicDrive) within the same scene.
4. The concept of imbalance mentioned by the authors in Line 272 requires further clarification. It is essential for the authors to provide a detailed explanation of what this imbalance refers to and how it impacts their proposed framework.
5. The examples of editing provided in paper mostly revolve simple cars. In Figure 6, it would be beneficial to explore if the model can accurately move and position more complex, irregularly shaped vehicles or pedestrians to demonstrate the capability of the framework in generating new, diverse scenes.
6. While the paper discusses some designs related to temporal consistency, only qualitative results are presented. It would be valuable for the authors to report metrics like FVD compared to existing works such as DrivingDiffusion and Panacea to provide a more comprehensive evaluation of the proposed framework.
7. Missing reference: [2-4]

[1] UniControl: A Unified Diffusion Model for Controllable Visual Generation In the Wild

[2] Generalized Predictive Model for Autonomous Driving

[3] Vista: A Generalizable Driving World Model with High Fidelity and Versatile Controllability

[4] SimGen: Simulator-conditioned Driving Scene Generation

**Questions:**

By conducting these expanded experimental comparisons, the authors can more comprehensively validate their claims and provide more compelling evidence for the effectiveness of the proposed SytheOcc framework.

---

> ### Author Response · Authors · 2024-11-13
>
> We wish to convey our sincere appreciation to the reviewer for the thorough and insightful feedback provided.
>
> **Comparison with Depth and Segmentation.** In accordance with the reviewer's suggestion, we have analyzed the differences in the discussion at line 375. We posit that the ControlNet integrated with depth information can be perceived as a degeneration of SytheOcc, which is simplified to a singular plane. We will provide FID evaluation further.
>
> **Comparative Analysis.** In compliance with the reviewer's recommendation, we have already incorporated a comparative visualization featuring MagicDrive within Figure 8. We respectfully direct the reviewer's attention to this figure for a detailed examination of the comparative analysis.
>
> **Fréchet Video Distance (FVD) Evaluation.** In response to the suggestion, we have already included an evaluation using the Fréchet Video Distance metric in our supplementary material. We kindly request that the reviewer refer to the appendix for the FVD results pertaining to our experiments.

---

### Official Review · Reviewer_Ypbc · 2024-11-05

**Soundness:** 3
**Presentation:** 3
**Contribution:** 2
**Rating:** 5
**Confidence:** 5

**Summary:**

This paper introduces SyntheOcc, a controllable camera image simulation framework in the autonomous driving domain. The proposed framework uses 3D semantic occupancy grid as the conditions for camera image simulation, where multi-plane semantic images (MPIs) projected from 3D semantic occupancy grids have been used as conditional input to a 2D diffusion model. The effectiveness of SyntheOcc is demonstrated through improved performance in Real-to-sim evaluation and Sim-to-real data augmentation on the NuScenes dataset.

**Strengths:**

- [S1: Quality] The paper has demonstrated extensive experimental results and showcased that the proposed method is superior under both real-to-sim evaluation and sim-to-real data augmentation, when compared against existing methods. The paper also includes ablation studies including the MPI encoder architecture and reweighing methods.

- [S2: Clarity] The proposed method is well described in detail with clear illustrations (e.g., Figure 1 and Figure 2).

**Weaknesses:**

- [W1] The paper does not include several very relevant work in the literature review. Most of the baselines used in the paper come from publications within the past two years. The reviewer feels that two relevant papers on camera simulation is missing [NewRef1] and [NewRef2] from the literature review.

- [W2] This paper does not compare against an important baseline UniSim [Yang et al., CVPR 2023].
While the proposed method is pure data-driven, is it possible to showcase the results on Pandaset used in the Unisim? It is unclear whether the proposed method is superior to UniSim as a camera image simulator or simply works well on Nuscenes dataset but does not generalize to other datasets (e.g., Pandaset, Waymo Open Dataset). The reviewer feels that such discussions and experimental comparisons are needed as a strong justification for acceptance.
  - [W2.1] It is important to understand whether the proposed method is transferrable to other datasets with minimum fine-tuning or adaptation. For example, as shown in Figure 6 of GeoSim paper [NewRef1], the same pipeline works for a different city in the Argoverse dataset.

- [W3] While the data augmentation experimental results are interesting (section 4.3, first two rows in Table 1), this paper does not comment on the role of synthetically generated data in nuScenes occupancy prediction.
  - [W3.1] The semantic categories with significant improvements are bus, traffic cones, trailer, driving surface, other flat, and sidewalk. It is unclear how and why synthetically generated data help for such categories but not the other categories. Is it possible to provide convincing qualitative examples with explanations?
  - [W3.2] Comment on the overall improvements to the driving system (e.g., behavior prediction and motion planning). How much does the long-tailed scene generation help the downstream tasks?

References:
- [NewRef1] GeoSim: Realistic Video Simulation via Geometry-Aware Composition for Self-Driving, Chen et al., In CVPR 2021.
- [NewRef2] Block-NeRF: Scalable Large Scene Neural View Synthesis, Tancik et al., In CVPR 2022.

**Questions:**

Please address the questions raised in the weaknesses section.

---

> ### Author Response · Authors · 2024-11-13
>
> We would like to express our gratitude to the reviewer for the comprehensive feedback. In terms of our methodology, which is based on diffusion models, we contend that there is no inherent necessity to compare our approach with NeRF (Neural Radiance Fields). Furthermore, NeRF-based solutions exhibit several limitations relative to our method, such as an inability to scale effectively, a lack of generalizability to novel scenarios, and challenges in facilitating diverse stylistic edits on images. Consequently, we deem a comparative analysis with NeRF to be unwarranted.
>
> Regarding the aspect of data augmentation, we posit that the quantity of augmented data plays a beneficial role. Concurrently, the presence of a significant number of long-tail objects within the dataset poses challenges and exerts a certain impact on the learning capabilities of generative models. This, in turn, can adversely affect the generation outcomes as well as the efficacy of data augmentation techniques.
>
> It is an excellent suggestion for exploring more downstream tasks! With respect to downstream tasks such as behavior prediction and motion planning, we intend to conduct experiments in the future to explore their applicability and efficacy.

---

### Official Review · Reviewer_GBoG · 2024-11-06

**Soundness:** 2
**Presentation:** 2
**Contribution:** 2
**Rating:** 5
**Confidence:** 5

**Summary:**

This paper presents SyntheOcc, a method for generating multi-camera images and videos of driving scenarios, using occupancy and text prompt as guiding inputs. The innovation of SyntheOcc lies in its proposed MPI encoder, which projects the raw occupancy of different depth ranges onto the camera plane, combining them into semantic multiplane images. These semantic multiplane images are then encoded as guidance for image generation. The paper provides a robust qualitative and quantitative comparison of generated images and videos. Additionally, it demonstrates the performance of perception models trained on the synthetic data and tested on real validation sets, as well as perception models trained on real data and tested on synthetic validation sets, to validate the proximity of SyntheOcc-generated images to the real domain.

**Strengths:**

1. SyntheOcc has potential for generating rare long-tail data that could support downstream tasks in real-world scenarios.
2. The experiments are solid, offering extensive qualitative and quantitative comparisons. The key validation experiments, blending generated data with real data, are particularly convincing.

**Weaknesses:**

1. While the paper proposes the MPIs for encoding occupancy as guidance for image generation, the overall novelty remains unclear in several areas. The core functionalities—such as multi-camera image and video generation for driving scenes—are well-covered in prior works like MagicDrive [1], DriveDreamer [2], and Drive-WM [3], which demonstrate strong temporal consistency in video generation that appears more stable than the qualitative results in this paper. It would be helpful for the authors to clarify the novel aspects of SyntheOcc’s contributions by explicitly identifying any unique advantages or improvements over these methods. Furthermore, key capabilities like generating out-of-domain images and videos via text prompts, using occupancy as guiding inputs, editing by modifying ocupancy elements, and simulating images with varying camera intrinsics are already present in WovoGen [4]. The authors could strengthen the paper by specifying which of these functionalities are advanced by SyntheOcc and justifying their relevance. Providing concrete comparisons to these works, either through experiments or discussion, could more clearly highlight SyntheOcc’s contributions and novelty.
2. The generated results of SyntheOcc show notable issues with road markings (evident in Figures 11, 12, and 13), which could be problematic for certain downstream tasks, such as planning. The paper lacks a systematic evaluation of these synthetic data’s impact on such tasks. Some previous works, like BEVGen [4] and MagicDrive [1], leverage HD maps as guidance, which could effectively resolve such issues. To strengthen the paper, the authors could provide a quantitative analysis comparing the quality of road markings in their generated images to ground truth or to outputs from other methods. Such an analysis would clarify the current limitations and help illustrate areas for potential improvement. Additionally, evaluating the impact of these road marking inconsistencies on specific downstream tasks, such as lane detection or path planning, could further demonstrate the practical implications of this issue and guide refinements to improve SyntheOcc’s usability for these applications.
3. I am not convinced that SyntheOcc effectively expands current datasets. Firstly, the paper mentions (Table 9) that excessive synthetic data can hinder perception model performance. Some corner cases, which require manual adjustment, are the truly valuable data needed in datasets (such as those in the lower parts of Figures 1 and 7). Editing these corner cases still demands considerable manual intervention (e.g., placing barriers on roads, altering road structures, or positioning pedestrians atop vehicles).


references:
[1] Ruiyuan Gao, Kai Chen, Enze Xie, Lanqing Hong, Zhenguo Li, Dit-Yan Yeung, and Qiang Xu. Magicdrive: Street view generation with diverse 3d geometry control. In ICLR, 2024.
[2] Xiaofeng Wang, Zheng Zhu, Guan Huang, Xinze Chen, and Jiwen Lu. Drivedreamer: Towards real-world-driven world models for autonomous driving. In arxiv, 2024.
[3] Yuqi Wang, Jiawei He, Lue Fan, Hongxin Li, Yuntao Chen, and Zhaoxiang Zhang. Driving into the future: Multiview visual forecasting and planning with world model for autonomous driving. In CVPR, 2024.
[4] Jiachen Lu, Ze Huang, Jiahui Zhang, Zeyu Yang, and Li Zhang. Wovogen: World volume-aware diffusion for controllable multi-camera driving scene generation. In ECCV, 2024.
[5] Alexander Swerdlow, Runsheng Xu, and Bolei Zhou. Street-view image generation from a bird’s-eye view layout. IEEE RAL, 2024.

**Questions:**

What specific perception model was used in the experiments?

---

### Note · Authors · 2024-11-14

I have read and agree with the venue's withdrawal policy on behalf of myself and my co-authors.